# IMPROVING OUT-OF-DISTRIBUTION GENERALIZATION BY MIMICKING THE HUMAN VISUAL DIET

## ABSTRACT

Human visual experience is markedly different from the large-scale computer vision datasets consisting of internet images. Babies densely sample a few $3D$ scenes with diverse variations such as object viewpoints or illuminations, while datasets like ImageNet contain one single snapshot from millions of 3D scenes. We investigated how these differences in input data composition (*i.e.,* visual diet) impact the Out-Of-Distribution (OOD) generalization capabilities of a visual system. Training models on a dataset mimicking attributes of the human-like visual diet improved generalization to OOD lighting, material, and viewpoint changes by up to $18\%$. This observation held despite the fact that the models were trained on $1,000$-fold less training data. Furthermore, when trained on purely synthetic data and tested on natural images, incorporating these visual diet attributes in the training dataset improved OOD generalization by $17\%$. These experiments are enabled by our newly proposed benchmark—the Human Visual Diet (HVD) dataset, and a new model (Human Diet Network) designed to leverage the attributes of a human-like visual diet. These findings highlight a critical problem in modern day Artificial Intelligence—building better datasets requires thinking beyond dataset size and rather focus on improving data composition. All data and source code will be made available upon publication.

## 1 INTRODUCTION

The development of the visual system is intricately tied to the visual experiences encountered from infancy (Kandel et al. (2000); Kreiman (2021); Arcaro et al. (2017); Hubel & Wiesel (1964); Daw & Wyatt (1976); Wood & Wood (2018; 2022); Bambach et al. (2018); Lee et al. (2021)). Growing evidence highlights the importance of visual experience (Smith & Slone (2017); Wood & Wood (2018; 2022); Sheybani et al. (2023; 2024); Tsotsos (1992); Tsotsos et al. (2019)). These visual experiences are constrained by the statistics of natural scenes (Simoncelli & Olshausen (2001)), resulting in data significantly different from large-scale datasets used in computer vision.

**Fig. 1**a illustrates two such differences. First, children learn from the physical space they occupy—a few 3D scenes and objects viewed under diverse real-world transformations including viewpoints, lighting, object textures, and occlusions. Second, children always view objects in the context of their surroundings. We refer to these as *real-world transformational diversity (RWTD)* and *scene context*, respectively. Here we investigate how these differences in input data composition impact Out-Of-Distribution (OOD) generalization performance.

Here we show that incorporating these visual diet attributes improves generalization. Models trained with a human-like visual diet achieve up to 18% improvement on OOD lighting, materials, and viewpoint changes. Training with such data outperforms training models on 1,000-fold larger internet datasets. Furthermore, when trained on synthetic images and tested on natural images, incorporating attributes of the human visual diet improved OOD generalization performance by up to 17%. These experiments are enabled by two key technical contributions. First, we introduce the **Human Visual Diet (HVD)** dataset, which mimics the input data during visual development and contains both transformational diversity and scene context (Sheybani et al. (2023); Smith & Slone (2017)) as shown in **Fig. 2**. Second, we propose the **Human Diet Network (HDNet)**—a model designed to leverage the attributes present in HVD (**Fig. 1**c). HDNet exploits transformational diversity by employing a contrastive loss over real-world transformations (lighting, material, $3D$ viewpoint changes), and

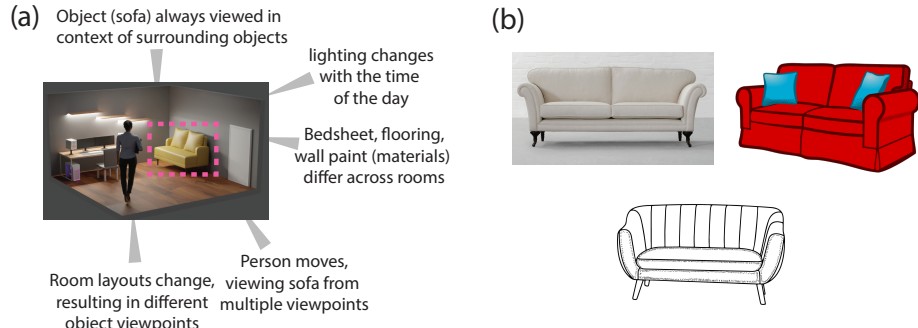

Figure 1: **Comparing the input data composition for humans and machines.** (a) Humans view the sofa in the context of its surroundings. Furthermore, the sofa is viewed under a variety of real-world transformations including variations in object viewpoints, changes to room lighting depending the time of the day, and material variations in the form of upholstery. (b) Both scene context and real-world transformational diversity (RWTD) are missing in internet scraped images of sofas.

uses a two-stream architecture to jointly reason over target and scene context to perform context aware visual recognition. To summarize, our work has three main contributions:

- We present three new benchmarks for measuring OOD generalization across disentangled, real-world transformations in lighting, materials, and viewpoint changes: the Human Visual Diet (HVD) dataset, Semantic-iLab dataset, and the Syn2Real dataset.
- We assess generalization capabilities of multiple computer vision architectures and domain generalization approaches on disentangled, semantic OOD shifts in these benchmarks.
- We show that incorporating real-world transformational diversity (RWTD) and scene context improves OOD generalization by large margins (as high as 17-18%), and present a new architecture, Human Diet Network (HDNet), designed to leverage these attributes.

## 2 RELATED WORK

Out-of-Distribution (OOD) generalization continues to be the Achilles heel of modern AI (Engstrom et al. (2018); Chaman & Dokmanic (2021); Zhang (2019)). Failure modes include OOD rotations and translations (Engstrom et al. (2018); Chaman & Dokmanic (2021); Zhang (2019)), real-world transformations including 3D viewpoints (Barbu et al. (2019); Liu et al. (2018); Zeng et al. (2019); Madan et al. (2021c); Sakai et al. (2022); Zheng et al. (2023)), changes in lighting (Madan et al. (2021c); Beery et al. (2018); Zhang et al. (2021)), and color changes (Joshi et al. (2019); Shamsabadi et al. (2020)), among other transformations.

Existing approaches to counter this generalization gap include—specialized architectures (Shahtalebi et al. (2021); Sun & Saenko (2016); Arjovsky et al. (2019); Kim et al. (2021); Vedantam et al. (2021); Krueger et al. (2021); Blanchard et al. (2017a)), novel pre-processing and data augmentation strategies (Yun et al. (2019); Hendrycks et al. (2019); Zhang et al. (2017); Madan et al. (2021b;a)), and generative modeling (Ilse et al. (2020); Wang et al. (2020)), among others. Lately, investigators introduced billion scale datasets like LAION-5B (Schuhmann et al. (2022)) and IG-1B Targeted (Yalniz et al. (2019)) hoping that they will leave little out of the distribution. However, despite progress, OOD samples remain an unsolved problem (Radford et al. (2021); Wortsman et al. (2022); Pham et al.). We introduce a new dataset and model inspired by the human visual diet.

Some recent work has emphasized the importance of training with more human-like data ( Bambach et al. (2018); Lee et al. (2021); Wood & Wood (2018; 2022)). These efforts include incorporating scene context (Zhang et al. (2020)), temporal structure (Sheybani et al. (2024)), binocular vision (Orhan et al. (2020); Orhan & Lake (2024)), and goal-directed/active sampling (Tsotsos (1992); Tsotsos et al. (2019); Bajcsy et al. (2018); Bajcsy (1988); Pelgrim et al. (2024)), among others. Our work extends these efforts to Out-of-Distribution generalization.

(a) Human Visual Diet (HVD) Dataset

Target (Sofa) viewed in
context of surrouding objects

Lighting Change

Material Change

Viewpoint Change

(b) Semantic-iLab Dataset

Original Image    Lighting Change    Material Change    Viewpoint Change

(c) Syn2Real Test Dataset

HVD (train images)

ScanNet (test image)

Figure 2: **Datasets with real-world transformations.** (a) Sample images from the Human Visual Diet (HVD) dataset. We created 15 photo-realistic domains with 3, disentangled real-world transformations—lighting, material, and viewpoint changes. Each 3D scene was created by reconstructing an existing ScanNet (Dai et al. (2017)) scene using the OpenRooms framework (Li et al. (2020b)), followed by introducing controlled changes in scene parameters before rendering. (b) Sample images from the Semantic-iLab dataset. We modified the iLab dataset (Borji et al. (2016)), augmenting images with changes in lighting and material by modifying the white balance and using AdaIN-based style transfer (Huang & Belongie (2017a)). (c) Sample images from the Syn2Real benchmark. HVD training images (left) and ScanNet testing images (right) show the same 3D scene. Models are trained on the purely synthetic HVD images, and tested on the natural ScanNet images.

# 3 DATASETS WITH CONTROLLED VARIATIONS IN LIGHTING, MATERIALS AND VIEWPOINTS

We present three new benchmarks for measuring OOD generalization across real-world transformations in lighting, materials, and viewpoint changes.

### 3.1 HUMAN VISUAL DIET (HVD) DATASET

$3D$ scenes from ScanNet (n=1,288) (Dai et al. (2017)) were reconstructed using the OpenRooms framework (Li et al. (2020a;b)), and 15 photo-realistic domains were constructed with these scenes by introducing 3 real-world transformations—lighting, material, and viewpoint changes. For each domain, $19,800$ images were rendered resulting in a total of $300,000$ images containing 1 million object instances with controlled variations in lighting, materials, and viewpoints (**Fig. 2a**).

**Light shift domains:** Outdoor lighting was controlled using 250 High Dynamic Range (HDR) environment maps from the Laval Outdoor HDR Dataset ( Hold-Geoffroy et al. (2019)) and OpenRooms. These were split into 5 sets of 50 each to create 5 light shift domains. We split the HSV color space into chunks of disjoint hue values. Each domain sampled indoor light color and intensity from one chunk. One domain was held out for testing (OOD Light), and never used for training (see sample images in **Fig. S1**).

**Material shift domains:** 250 procedural materials from Adobe Substances were used, including different types of wood, fabrics, floor and wall tiles, and metals, among others. These were split into sets of 50 materials each to create 5 different material domains. For each material domain, one of these 50 materials were randomly assigned to each scene object. One domain was held out for testing (OOD Materials), and never used for training (see sample images in **Fig. S2**).

**Viewpoint shift domains:** Disjoint viewpoint domains were constructed by changing the height at which the camera focuses, *i.e.,* the zenith angle. Five viewpoint domains were constructed, and one was held out for testing (OOD Viewpoints, see sample images in **Fig. S3**).

### 3.2 SEMANTIC-ILAB DATASET

Images from iLab (Borji et al. (2016)) were modified to create a natural image dataset with variations in lighting, material and viewpoints (**Fig. 2b**). iLab contains objects from 15 categories placed on a turntable and photographed from varied viewpoints. Fist, a foreground detector was used to extract the object. Then, material variations were implemented using AdaIN-based style transfer (Huang & Belongie (2017b)) on these object masks and the style transferred object was overlaid onto the original background. Lighting changes were simulated by modifying the white balance. Unlike HVD, this dataset does not contain scene context (see **Sec. B** for more details).

### 3.3 SYN2REAL DATASET: NATURAL IMAGE TEST SET FROM SCANNET

The Syn2Real dataset is composed of a test set of natural images from the ScanNet dataset, and a training set of only synthetic images from HVD. The natural image test set was created by annotating images from ScanNet ( Dai et al. (2017)). To capture distinct images, one frame was sampled every 100 frames from ScanNet's raw video footage. These frames were then annotated using LabelMe (see **Sec. H** for further details).

## 4 HUMAN DIET NETWORK (HDNET)

A schematic of HDNet is shown in **Fig 3**. There are two main components to this model. First, a two-stream network inspired by the eccentricity dependence of human vision that jointly reasons over target object and context. Second, a contrastive loss over real world transformations.

**Two-stream network inspired by human vision:** Given the training dataset $D = \{x_i, y_i\}_{i=1}^{n}$, HDNet is presented with an image $x_i$ with multiple objects and the bounding box for a single target object location. The target ($I_{i,t}$) is obtained by cropping the input image $x_i$ to the bounding box whereas $I_{i,c}$ covers the entire contextual area of the image $x_i$. $y_i$ is the ground truth class label for $I_{i,t}$. The first stream processes only the target object ($I_t$, $224 \times 224$) and outputs $y_t$, while the second stream processes the periphery ($I_c$, $224 \times 224$) and outputs $y_{t,c}$. Based on the confidence in the prediction $y_t$ (denoted $p$), HDNet computes a confidence-weighted average of $y_t$ and $y_{t,c}$ to get the final prediction $y_p$. If the model makes a confident prediction with the object only, it overrules the context reasoning stage.

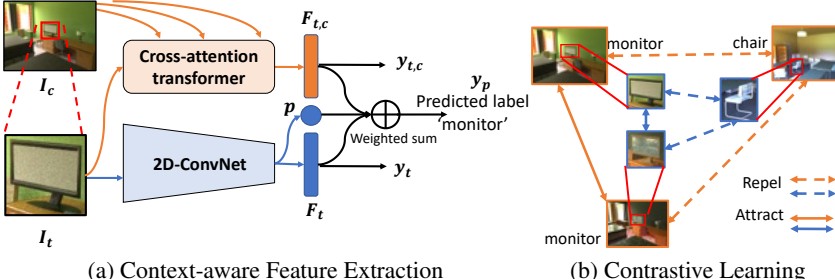

(a) Context-aware Feature Extraction    (b) Contrastive Learning

Figure 3: **Architecture overview for the Human Diet Network(HDNet)**. (a) Modular steps carried out by HDNet in context-aware object recognition. HDNet consists of 3 modules: feature extraction, integration of context and target information, and confidence-modulated classification. HDNet takes the cropped target object $I_t$ and the entire context image $I_c$ as inputs and extracts their respective features. These feature maps are tokenized and information from the two streams is integrated over multiple cross-attention layers. HDNet also estimates a confidence score $p$ for recognition using the target object features alone, which is used to modulate the contributions of $F_t$ and $F_{t,c}$ in the final weighted prediction $y_p$. (b) To help HDNet learn generic representations across domains, we introduce contrastive learning on the context-modulated object representations $F_{t,c}$ in the embedding space. Target and context representations for objects of the same category are enforced to attract each other, while those from different categories are enforced to repel. Pairs for contrastive learning are generated using various lighting, material, or viewpoint shifts (**Sec. 3.1**).

**Contrastive loss over real-world transformations:** HDNet builds on the supervised contrastive learning loss (Khosla et al. (2020))—samples from the same object category (but different lighting, material, or viewpoint) serve as positive pairs, while samples from different object categories serve as negative pairs. Consider a batch of $N$ data and label pairs $\{x_k, y_k\}_{k=1}^{N}$. The corresponding multiview batch consists of $2N$ pairs of domain-shifted images constructed by modifying the lighting, materials or viewpoints of objects in the batch. $\{\tilde{x}_l, \tilde{y}_l\}_{l=1}^{2N}$, where $\tilde{x}_{2k}$ and $\tilde{x}_{2k-1}$ are two views created with random semantic domain shifts of $x_k (k = 1, ..., N)$ and $\tilde{y}_{2k} = \tilde{y}_{2k-1} = \tilde{y}_k$. The domain shifts are randomly selected from a set of HVD domains specified during training. For example, if $x_k$ is from a material domain, $\tilde{x}_{2k}$ and $\tilde{x}_{2k-1}$ would be images from the same 3D scene but with different materials.

Within a multiviewed batch, let $m \in M := \{1, ..., 2N\}$ be the index of an arbitrary domain shifted sample. Let $j(m)$ be the index of the other domain shifted samples originating from the same source samples belonging to the same object category, also known as the positive. Then $A(m) := M\backslash\{m\}$ refers to the rest of indices in $M$ except for $m$ itself. Hence, we can also define $P(m) := \{p \in A(m) : \tilde{y}_p = \tilde{y}_m\}$ as the collection of indices of all positives in the multiviewed batch distinct from $m$. $|P(m)|$ is the cardinality. The supervised contrastive learning loss is:

$$L_{contrast} = \sum_{m \in M} L_m = \sum_{m \in M} \frac{-1}{|P(m)|} \sum_{p \in P(m)} \log \frac{\exp(z_m \cdot z_p/\tau)}{\sum_{a \in A(m)} \exp(z_m \cdot z_a/\tau)} \quad (1)$$

Here, $z_m$ refers to the context-dependent object features $F_{m,t,c}$ on $\tilde{x}_m$ after L2 normalization. This design encourages HDNet to attract the objects and relevant context from the same category, and repel the objects and irrelevant context from different categories. HDNet is trained end-to-end with contrative loss alongside three three cross-entropy losses proposed in context-aware CRTNet architecture Bomatter et al. (2021). These include cross-entropy losses. First, loss w.r.t. the confidence-weighted prediction $y_p$ denoted $L_p$, which allows the model to increase the confidence value $p$ for samples where the prediction based on target alone tends to be correct. Second, w.r.t. $y_t$, denoted $L_t$. (iii) Finally, w.r.t. $y_{t,c}$, denoted $L_{c,t}$. This disentangled objective function ensures strong learning signals for all parts of the architecture irrespective of the value of $p$. Thus, HDNEt is trained end to end with:

$$L_{HDNet} = \alpha L_{contrast} + L_p + L_t + L_{c,t} \quad (2)$$

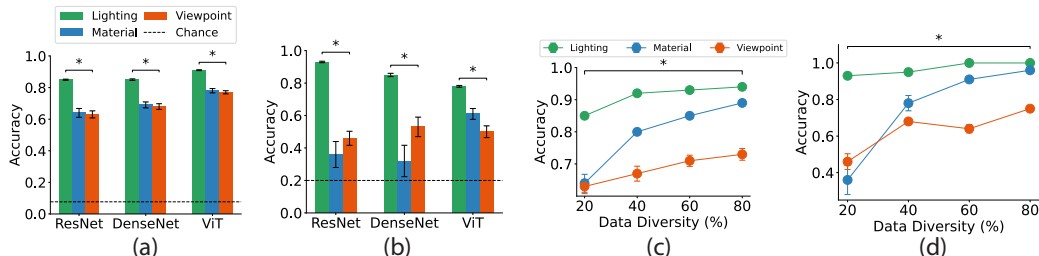

Figure 4: **Real-world transformational diversity significantly improved generalization for all OOD transformations and architectures.** (a) Models trained with low transformational diversity and minimal context struggled to generalize across real-world OOD transformations (especially material and viewpoint changes). Y-axis reports the top-1 classification accuracy for ResNet, DenseNet and ViT models trained on the HVD dataset (b) ResNet, DenseNet and ViT models trained on the Semantic-iLab dataset also struggled to generalize to OOD transformations. (c) Generalization improved significantly as real-world transformational diversity (RWTD) is increased. Y-axis reports accuracy for a HDNet model trained on the HVD dataset. This held true for all OOD transformations. (d) Generalization performance also improved for a ResNet model trained on the Semantic-ilab dataset. An ∗ represents statistical significance (two-sided t-test). The colors green, blue, and red represent performance on OOD lighting, OOD materials, and OOD viewpoints respectively.

Similar to past work (Bomatter et al. (2021); Zhang et al. (2020)), we set $\tau = 0.1, \alpha = 0.5$ to balance the supervision from constrastive learning and the classification loss. Learning rate was set to 0.0001 and all models were trained the Adam optimizer.

## 5 RESULTS

We present findings demonstrating the effectiveness of training models with two key attributes of the human visual diet:—Real-World Transformational Diversity (RWTD) and Scene Context. To begin, Sec. 5.1 establishes a lower baseline by benchmarking generalization capabilities of conventional vision models trained with low RWTD and minimal scene context. Building further, Sec. 5.2 and Sec. 5.3 respectively show that incorporating real-world transformation diversity and scene context into the training data improves generalization significantly. Sec. 5.4 presents an extreme test of our hypothesis, showing that models trained with a human-like visual data outperform models trained on a 1000x-fold larger internet-scraped dataset. Finally, as a real-world litmus test, Sec. 5.5 shows that a human-like visual diet leads to significant improvement in generalizing from synthetic images from HVD to natural images from ScanNet.

For all experiments, one domain per transformation (light, material, viewpoint) was held out as the OOD test set and never used for training. As Real-World Transformational Diversity (RWTD) was increased from 1 to 4 domains (20% to 80% RWTD), the number of images sampled per domain were reduced. This ensured a fixed training dataset size. All models were pre-trained on ImageNet. Additional details on hyperparameters and baseline models are provided in Sec. I and Sec. J.

### 5.1 MODELS TRAINED WITH LOW DIVERSITY AND MINIMAL CONTEXT STRUGGLE TO GENERALIZE

We started by estimating a lower baseline by training models on a dataset analogous to internet-scraped datasets like ImageNet ( Deng et al. (2009)). For this purpose, we evaluated generalization performance of three common vision models: ResNet He et al. (2016), DenseNet ( Huang et al. (2017)), and ViT ( Dosovitskiy et al. (2020)) when trained with low real-world transformational diversity (RWTD) and minimal scene context. Specifically, these models were trained with data from only 1 domain (low RWTD) and cropping images to show only the target object and testing on other domains (minimal context). These results are reported in **Fig. 4a,b**.

For HVD (**Fig. 4a**), ResNet generalized better across lighting changes (green) than material changes (blue, two-sided t-test, $p < 10^{-5}$) or viewpoint changes (red, two-sided t-test, $p < 10^{-6}$). There is

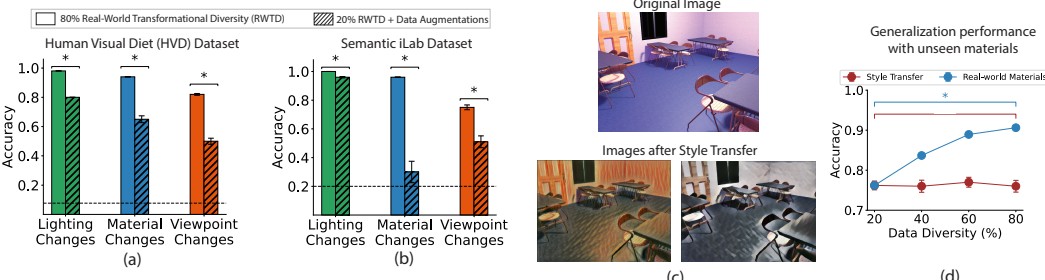

Figure 5: **Data post-processing does not match gains from collecting data mimicking the human visual diet**. (a),(b) Models trained with 80% real-world transformational diversity (RWTD) (solid bars) outperform those trained with 20% RWTD + traditional data augmentation (striped bars) for all transformations (lighting (green), material (blue), and viewpoint (red)) across both HVD (a) and Semantic-iLab (b) datasets. Number of images is held constant in these experiments. Format and conventions as in **Fig. 4ab**. (c) Sample images (bottom) from style transfer domains created using AdaIn (Huang & Belongie (2017a)) based on the original image on top. (d) Models trained on style transfer domains (red) do not generalize as well as those trained with material diversity (blue). The increase in OOD performance for real-world materials is statistically significant (blue, significance denoted by '*'). However, there is no statistically significant increase in performance when the model is tranied with additional style-transfer domains (red, without an '*').

ample room for improvement, especially when tested on OOD material and viewpoints. Similar conclusions can be drawn for DenseNet and ViT. The qualitative conclusions were similar for Semantic-iLab (**Fig. 4**b)—ResNet generalized better across OOD lighting (**Fig. 4**b, green) than OOD materials (blue, two-sided t-test, $p < 10^{-6}$) or OOD viewpoints (red, two-sided t-test, $p < 10^{-6}$). Furthermore, the degree of generalization for material and viewpoints was particularly low for Semantic-iLab. These conclusions on the Semantic-iLab dataset held true for DenseNet and ViT architectures as well. In sum, models trained with minimal diversity and minimal context showed only moderate generalization, especially struggling with material and viewpoint changes.

## 5.2 REAL-WORD TRANSFORMATIONAL DIVERSITY (RWTD) IMPROVES GENERALIZATION FOR ALL OOD TRANSFORMATIONS

For each transformation, we increased the amount of real-world transformational diversity (RWTD) that the models were exposed to by training them using samples from more domains from 1 domain (20%) to 4 domains (80%). Fig. 4 reports results for our proposed Human Diet Network (HDNet) trained with the HVD dataset, and for a ResNet model trained with the Semantic-iLab dataset as transformational diversity is increased. As described in **Sec. 3** and **Sec. 4**, HDNet was designed specifically to leverage the transformational diversity present in the HVD dataset.

OOD generalization improved approximately monotonically with transformational diversity for all three transformations in the HVD dataset (**Fig. 4**c). This improvement was significantly greater for OOD materials than for OOD lighting ($p < 10^{-4}$) and OOD viewpoints ($p < 10^{-4}$). Increased diversity improved generalization for a ResNet model trained on the Semantic-iLab dataset as well (**Fig. 4**d). As with HVD, improvement in generalization across the Semantic-iLab dataset was higher for unseen materials than for unseen lighting ($p < 10^{-3}$) and unseen viewpoints ($p < 10^{-6}$). Thus, with sufficient diversity, generalization to OOD lighting and materials reached almost ceiling levels. However, despite improvement, OOD viewpoints remained a challenge.

**RWTD outperforms data augmentation:** We compared the impact of training with real world transformations versus training with traditional data augmentation (Crops, Rotations, Contrast, and Solarize operations). **Fig. 5**a compares HDNet trained on HVD with 80% RWTD, and the same architecture trained with 20% RWTD+traditional data augmentation. RWTD outperformed data augmentation for all three transformations (two-sided t test, $p < 10^{-4}$). The same was true for a ResNet model trained with Semantic-iLab dataset (**Fig. 5(b)**). For additional details, see **Sec. G**.

| Transfor-
mation | AND
Mask | CAD | COR
AL | ERM | IRM | MTL | Self
Reg | VREx | Faster
RCNN | HDNet
(ours) |
|---|---|---|---|---|---|---|---|---|---|---|
| Lighting | 0.82 | 0.80 | 0.81 | 0.8 | 0.83 | 0.81 | 0.75 | 0.83 | 0.95 | **0.98** |
| Materials | 0.75 | 0.75 | 0.75 | 0.75 | 0.74 | 0.74 | 0.74 | 0.75 | 0.78 | **0.94** |
| Viewpoints | 0.75 | 0.77 | 0.79 | 0.78 | 0.76 | 0.79 | 0.76 | 0.78 | 0.65 | **0.83** |

Table 1: **Contextual information improves OOD generalization.** All models were trained with 4 HVD domains (80% RWTD) and tested on the 1 held-out domain for each of the three transformations. HDNet is benchmarked against specialized domain generalization (DG) baselines, and two context-aware baselines—a FasterRCNN model modified to do recognition, and CRTNet Bomatter et al. (2021). Similar to past works Gulrajani & Lopez-Paz (2020), ERM performed the best among the DG baselines. CRTNet was the best performing baseline, but HDNet outperformed all baselines on all three transformations. Bolded entries indicate the highest accuracy in each row.

| Semantic
Shift | Full
Context
($\sigma = 0$) | Less
Context
($\sigma = 25$) | Least
Context
($\sigma = 125$) |
|---|---|---|---|
| Lighting | **0.98 ± 0.001** | 0.96 ± 0.001 | 0.94 ± 0.001 |
| Material | **0.94 ± 0.002** | 0.88 ± 0.01 | 0.83 ± 0.006 |
| Viewpoints | **0.83 ± 0.006** | 0.77 ± 0.01 | 0.76 ± 0.01 |

Table 2: **Blurring scene context worsens generalization performance**. $\sigma$ is the standard deviation for the gaussian kernel applied to the image as a filter. Thus, blurring increases with $\sigma$, and is applied to both training and testing data. Similar to Table. 1, HDNet is trained on 4 domains and tested on the held-out OOD domain for each transformation. As blurring increases, the generalization performance of HDNet drops on all three OOD transformations (Lighting, Material and Viewpoints).

**RWTD outperforms generative AI:** Fig. 5(c) shows style-transfer domains constructed using the generative AI model AdaIn Huang & Belongie (2017a), an alternate approach to the rendered, photorealistic materials used in HVD. 5(d) shows generalization performance of HVD trained with real-world materials from HVD vs images from these style-transfer domains. The training dataset size was kept constant, and all models were tested on the same held-out OOD Materials domain. Unlike new material domains, new style transfer images did not improve generalization to OOD materials. Additional details are provided in **Sec. E.2**.

## 5.3 UTILIZING SCENE CONTEXT IMPROVES GENERALIZATION

We compared HDNet with a suite of baselines that do not utilize scene context. This includes domain generalization (DG) architectures—ANDMask (Shahtalebi et al. (2021)), CAD (Blanchard et al. (2017a)), CORAL (Sun & Saenko (2016)), MTL (Blanchard et al. (2017b)), Self-Reg (Kim et al. (2021)), VREx (Krueger et al. (2021)), IRM (Arjovsky et al. (2019)), and ERM (Gulrajani & Lopez-Paz (2020)). We also report comparisons with two context-aware models—a modified FasterRCNN model designed to perform visual recognition and the recent CRTNet (Bomatter et al. (2021)) — to the comparison. All models were trained with 80% Transformational Diversity, i.e., 4 training domains.

Table 1 reports the top-1 classification accuracy of HDNet compared with the above listed baselines. HDNet beat all baselines with statistical significance (two-sided t-test, $p < 0.05$) for all three transformations. The best performing baseline was another context-aware model—CRTNET Bomatter et al. (2021). The best performing DG approach was ERM, which was outperformed by CRTNet. In summary, approaches utilizing scene context (HDNet and CRTNet) outperformed all specialized DG approaches on all real-world transformations, and our proposed HDNet also outperformed the closest baseline (CRTNet).

**Removing scene context worsens generalization:** The results above show that incorporating scene context improves generalization. Additionally, **Table. 2** shows the impact of reducing scene context

| Real-World Transformation | Architecture | 1 Stream | 2 Stream |
|---|---|---|---|
| Lighting | ResNet | $0.85 \pm 0.004$ | $0.95 \pm 0.009^*$ |
| | ViT | $0.91 \pm 0.003$ | $0.97 \pm 0.007^*$ |
| | HDNet (Ours) | - | $\mathbf{0.98 \pm 0.001}$ |
| Materials | ResNet | $0.64 \pm 0.03$ | $0.83 \pm 0.008^*$ |
| | ViT | $0.78 \pm 0.01$ | $0.92 \pm 0.003^*$ |
| | HDNet (Ours) | - | $\mathbf{0.94 \pm 0.002}$ |
| Viewpoint | ResNet | $0.63 \pm 0.02$ | $0.72 \pm 0.009^*$ |
| | ViT | $0.77 \pm 0.01$ | $0.83 \pm 0.001^*$ |
| | HDNet (Ours) | - | $\mathbf{0.83 \pm 0.006}$ |

Table 3: **Modifying standard architectures to leverage scene context.** We proposed a methodology to modify standard architectures such that they can utilize scene context. Inspired by HDNet, a ResNet and a ViT model were modified to have two streams—one operating on the target, and the other one on the contextual information. This modification significantly improved generalization across all OOD transformations for both ResNet and ViT trained on the HVD dataset. All models were trained on 4 domains (80% RWTD) and tested on the held out domain for each transformation. Best performing model for each transformation has been bolded.

| Real World Transformation | Dino V2 | ResNet50 SWSL | ResNet18 SWSL | ResNext101 32x4d SWSL | ResNext101 32x16d SWSL | ResNext50 32x4d SWSL | HDNet (Ours) |
|---|---|---|---|---|---|---|---|
| Lighting | 0.94 | 0.9 | 0.88 | 0.93 | 0.93 | 0.91 | **0.98** |
| Materials | 0.79 | 0.73 | 0.67 | 0.77 | 0.79 | 0.74 | **0.94** |
| Viewpoints | 0.74 | 0.72 | 0.65 | 0.74 | 0.78 | 0.73 | **0.83** |

Table 4: **Our approach beats models trained with 1,000x more data.** HDNet was pre-trained on ImageNet and fine-tuned on data with both transformational diversity and scene context (4 HVD domains, full scene context). Baselines were pre-trained on 1,000-fold more data (IG-1B dataset), but fine-tuned on data not containing these two attributes (1 HVD domain, minimal scene context). HDNet beats all baselines by a large margin for all three transformations.

information by blurring the context using a Gaussian Blur. Performance dropped consistently for all three transformations as contextual information is reduced.

**Modifying existing architectures to leverage scene context:** We present a simple methodology to modify existing architectures (ResNet, ViT) such that they can leverage scene context. For ResNet, a two-stream version was made where each stream was a ResNet backbone. One stream operated on the target, and the other on the scene context. Output features from each stream were concatenated, and passed through a fully connected layer for classification. Two-stream ViT was analogous. In contrast, the one-stream architecture did not use scene context and operated on the target object alone. These modifications led to significantly improved performance (two-sided t test, $p < 0.05$), as shown in **Table 3**. Additional experiments on the role of scene context are presented in **Sec. E.1**.

### 5.4 HUMAN-LIKE VISUAL DIET BEATS BILLION-SCALE INTERNET-SCRAPED DATASETS

Next, we compared HDNet with visual recognition models trained with 1,000x more data (**Table. 4**). All models except HDNet were pre-trained on the IG-1B dataset Yalniz et al. (2019), and then fine-tuned on data with 20% RWTD and with object crops *i.e.,* low transformational diversity and minimal context. In comparison, HDNet was pre-trained on ImageNet and fine-tuned with data consisting of 80% RWTD and scene context *i.e.,* human-like visual diet. All models were fine-tuned on the same number of images. HDNet outperformed all billion-scale baselines by large margins despite being trained on 1000x less data (**Table. 4**, two-sided t-test, $p < 0.001$).

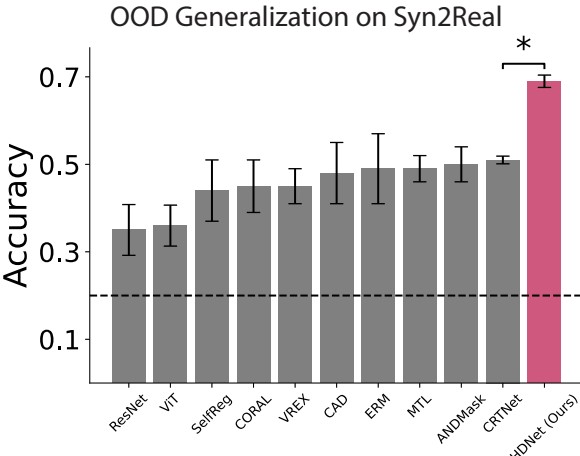

Figure 6: **Human-like visual diet enables improved generalization on the Syn2Real benchmark.** HDNet is compared against a suite of DG benchmarks, standard computer vision models, and the context-aware CRTNet model. All models are trained on purely synthetic images (HVD) and tested on natural images (ScanNet) with no pre-training on any natural images. HDNet beats all baselines by a large margin (18%), and with statistical significance.

### 5.5    HUMAN-LIKE VISUAL DIET ENABLES GENERALIZATION TO REAL-WORLD IMAGES

As a real litmus test, we tested the impact of a human-like visual diet on the Syn2Real benchmark *i.e.,* models were trained on purely synthetic images (from HVD) and tested on natural images from ScanNet. These models were not pretrained on ImageNet, and thus, had never seen any natural images. **Fig. 6** reports a comparison of HDNet is compared against a suite of DG benchmarks, standard computer vision models, and the context-aware CRTNet model. HDNet trained with RWTD and scene context achieved an accuracy of $0.69$, while the best baseline (CRTNet) trained without a human-like diet achieved an accuracy of $0.51$. Thus, incorporating these attributes into the training dataset enabled HDNet to generalize significantly well from a purely synthetic training data to a natural image test set (two-sided t-test, $p < 0.05$).

## 6    CONCLUSIONS

We investigated the impact of data composition on the out-of-distribution generalization capabilities of visual recognition models. Specifically, we demonstrated that incorporating two key components of the human visual diet—transformational diversity and scene context—improves generalization to OOD viewpoints, lighting, and material changes. Our contributions include three new benchmarks, and a novel architecture that models and leverages these human-like visual attributes. This work provides an approach complementary to existing directions on data augmentation and specialized domain generalization architectures.

While our results are promising, the human visual diet is complex and multifaceted, with several additional features like temporal information, egocentric views, embodiment, and goal-driven/active sampling that warrant further investigation. We hope that future datasets extending the Human Visual Diet (HVD) dataset introduced here can address these. Similarly, the Human Diet Network (HDNet) introduced here represents a promising first step in integrating human-like scene context, but is currently limited to only spatial context. We hope that future work can build architectures incorporating temporal context—such as motion and sequential dependency. In summary, this work opens new avenues for aligning biological and artificial vision systems, and advancing generalization in AI. Out-of-distribution generalization remains the Achilles' heel of modern AI, and we hope future research in these directions will lead to models that generalize as effortlessly as human vision.

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

## SUPPLEMENTARY MATERIALS

## A    HVD DOMAINS

### A.1    SAMPLE IMAGES FROM THE HVD DATASET

We present additional images from the HVD dataset. Each figure shows change in one scene parameter, while holding all others constant. In Fig. S1 we show images from two different light domains. Note that the first three rows in Fig S1 show different indoor lighting conditions controlled using indoor light color and intensity sampled from disjoint chunks of the HSV space. The last two rows show different outdoor lighting settings created by changing the environment maps. Similarly, Fig. S2 shows five different scenes from two training domains with a material shift. Fig. S3 shows viewpoint shifted domains.

## B    DETAILS ON THE CONSTRUCTION OF THE SEMANTIC ILAB DATASET

We show sample images from the *Semantic iLab* dataset in Fig. 2(b) created by modifying the existing iLab Borji et al. (2016) dataset. This is a multi-view dataset, and hence already contains viewpoint shifted variations of the same objects. We modify the dataset to also contain material and light shifts. To mimick light shift, we modified the white balance of the original images, as shown in Fig. 2(b). For material shifts, we first run a foreground detector on these objects using Google's Cloud Vision API. We also run style transfer on these images using AdaIn Huang & Belongie (2017b). Then, we overlay the style transferred image on to the object mask on the original image to mimick material shifts. Note that this is approximate, and does not model the physics of material transfer in the same way as our rendered HVD dataset which is far more photorealistic, as shown in Fig. S2. Material shifted *Semantic iLab* images are shown in Fig. 2(b). As the dataset is originally multi-view, we do not need to generate new viewpoints and can use images of a different viewpoint from the original dataset as shown in Fig. 2(b).

## C    DETAILS ON THE CONSTRUCTION OF THE SYN2REAL DATASET

Results are reported on a test subset of 350 test images which are not blurry and do not have significant clutter, and on a larger subset of 700 test images where clutter and minor blurring was allowed to achieve a bigger test set. The same procedure was also followed to hand-annotate $8,000$ training object instances from the HVD dataset to ensure there is no spurious impact of the annotation procedure on the performance of models when tested on ScanNet. We made three adaptations for these experiments. Firstly, as both ScanNet and ImageNet contain natural images and overlapping categories, we trained models from scratch to ensure pre-training does not interfere with our results. Thus, these models never saw any real-world images, not even ImageNet as they were not pretrained on those datasets. Secondly, we trained and tested models on overlapping classes between HVD and ScanNet. Finally, we used the LabelMe Wada (2018) software to manually annotate a test set from ScanNet and training set for the HVD dataset using the same procedure to make sure biases from the annotation procedure do not impact experiments. Thus, all models were trained purely on synthetic data from HVD and tested on only real-world natural image data from ScanNet as shown in **Fig. 2(c)**.

## D    DETAILS ON THE HUMAN DIET NETWORK

## E    ADDITIONAL EXPERIMENTS WITH REAL-WORLD TRANSFORMATIONAL DIVERSITY

### E.1    REAL-WORLD TRANSFORMATIONS OUTPERFORM TRADITIONAL DATA AUGMENTATION.

We investigated how real-world transformational diversity (RWTD) compares to traditional data augmentation strategies including 2D rotations, scaling, and changes in contrast. Models trained with a visual diet consisting of $80\%$ RWTD were reported in **Fig.3(e)**. We compared these with

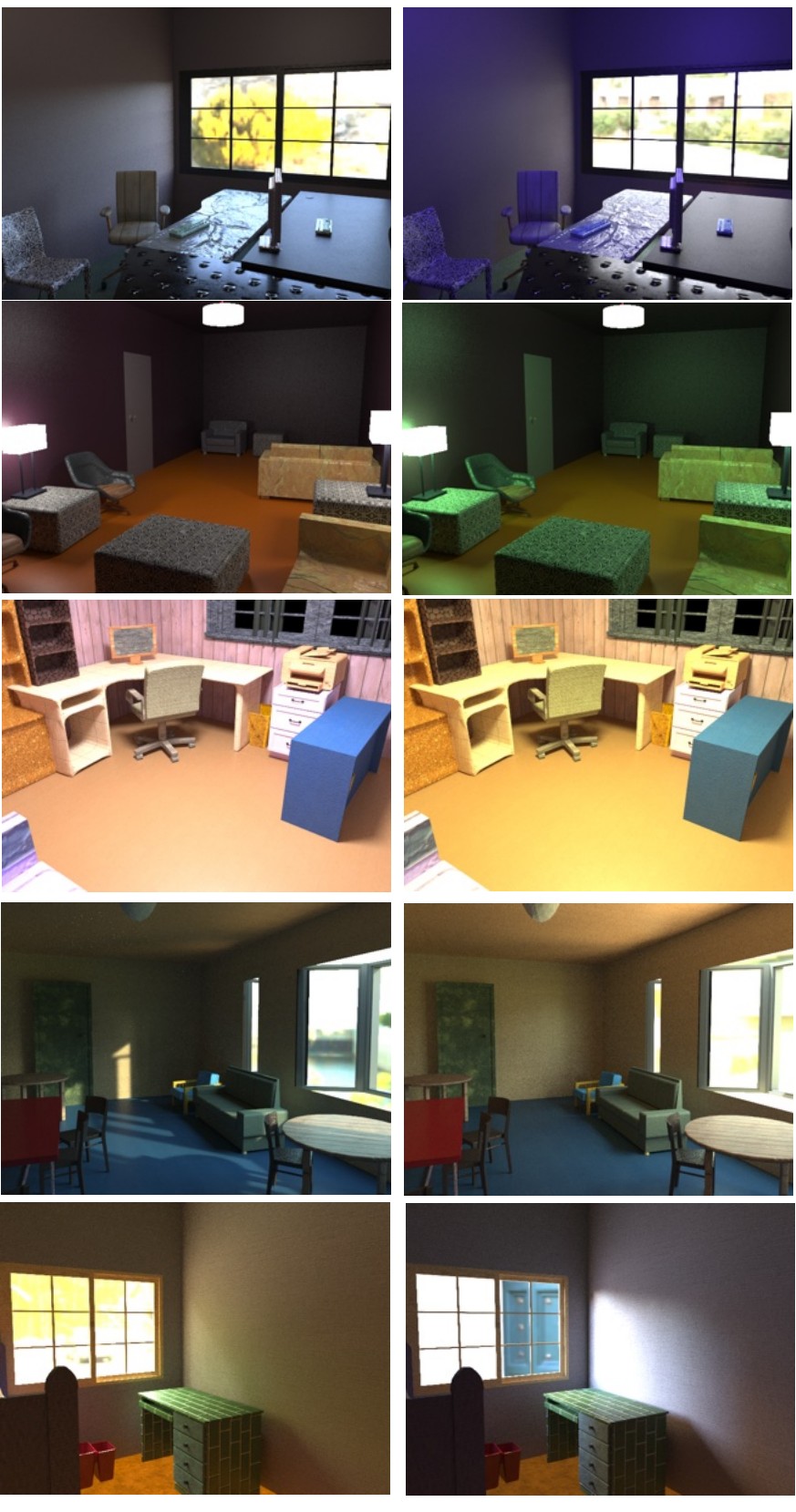

Figure S1: *Example images showing lighting tranformations.* We show paired images from different lighting transformation domains between the right and left column in each row. All other parameters held constant.

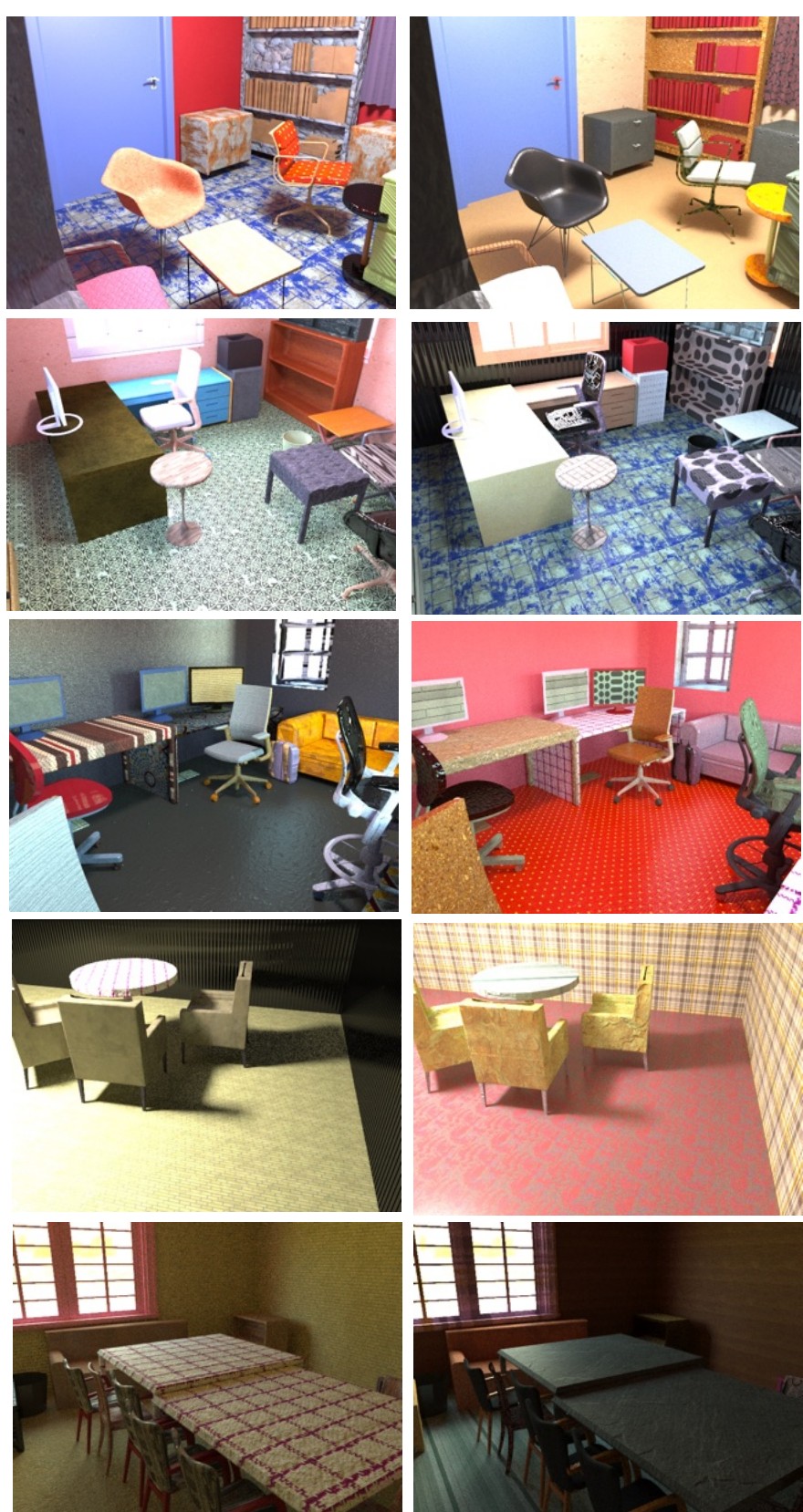

Figure S2: *Example images showing material tranformations.* We show paired images from different material transformation domains between the right and left column in each row. All other parameters held constant

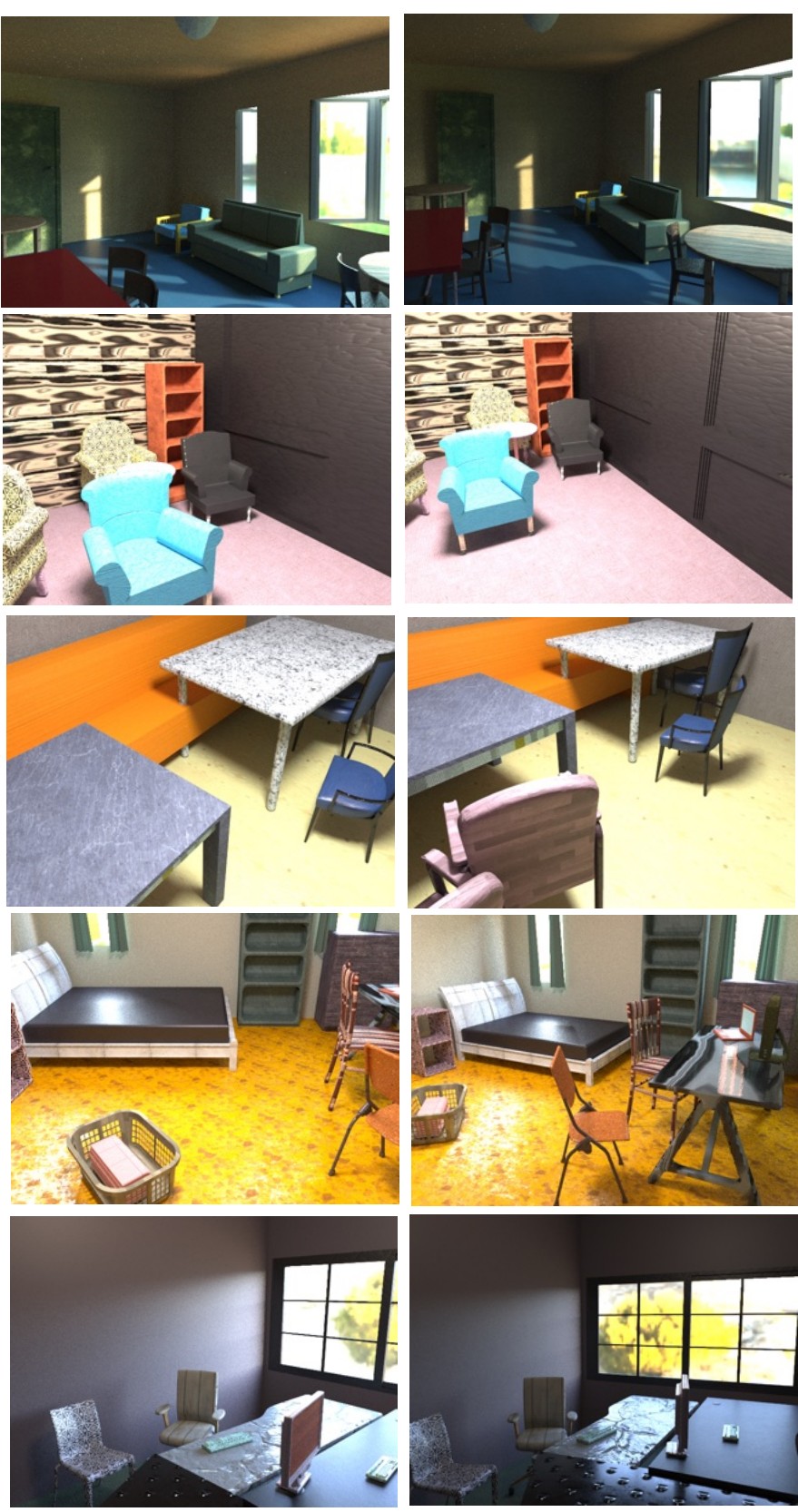

Figure S3: *Example images showing viewpoint tranformations.* We show paired images from different viewpoint transformation domains between the right and left column in each row. All other parameters held constant

models trained with a visual diet consisting of 20% RWTD + traditional augmentation. As before, all models were tested on unseen lighting, material, and viewpoint changes.

The number of training images was kept constant across all training scenarios to evaluate the quality of the training images rather than their quantity. Training set size equalization was achieved by sampling fewer images per domain in the 80% RTWD training set. For instance, for HVD experiments with unseen viewpoints we sampled $15,000$ training images per viewpoint domain to construct the training set with 20% RWTD + Data Augmentations. In comparison, we sampled only $3,750$ per viewpoint domain to construct the 80% RWTD training set. Thus, the initial sizes of the 80%RWTD and the 20%RWTD+Data Augmentation training sets was identical. However, due to data augmentations being stochastic the total number of unique images shown to models trained with data augmentations was much larger. Assuming a unique image was created by data augmentation in every epoch, over 50 epochs the dataset size would be 50 times larger with data augmentations.

Traditional data augmentation largely involves 2D affine operations (crops, rotations) or image-processing based methods (contrast, solarize) which are not necessarily representative of real-world transformations. In summary, the positive impact of a visual diet consisting of diverse lighting, material, and viewpoint changes (real-world transformational diversity) cannot be replicated by using traditional data augmentation applied to the dataset after data collection—diversity must be ensured at the data collection level.

### E.2 REAL-WORLD TRANSFORMATIONS OUTPERFORM AUGMENTATION WITH GENERATIVE AI.

Several existing works rely on increasing data diversity using AdaIn-based methods Huang & Belongie (2017a); Zhou et al. (2021). These style transfer methods change the colors in the image while retaining object boundaries, but do not modify materials explicitly as done in our HVD dataset. We evaluated how well models perform if diversity is increased using style transfer as opposed to material diversity. We started with one material domain, and created four additional domains using style transfer. Sample images of style transfer domains are shown in **Fig. 5(c)**. Corresponding images from the HVD dataset with real-world transformation in materials can be seen in **Fig. 2(a)**. The total number of domains (and images) created using style transfer was kept the same as the material domains in HVD. The only difference in the training data was that instead of four additional material domains, we have four additional style transfer domains. We compared models trained with these two different visual diets—one consisting of four material domains, and the other consisting of four style transfer domains. All models were then tested on the same held-out OOD Materials domain. Style transfer domains did not enable models to generalize to new materials as well as the material shift domains presented in HVD (**Fig. 5(d)**).

These experiments support the notion that in order to build visual recognition models that can generalize to unseen materials, it is important to explicitly increase diversity using additional materials at the time of training data collection. The impact of diverse materials cannot be replicated by using style transfer to augment the dataset after data collection.

### E.3 EACH INDIVIDUAL REAL-WORLD TRANSFORMATION IS HELPFUL

Some real-world transformations are easier to capture than others. For instance, capturing light changes during data collection might be significantly easier than collecting multiple possible room layouts, or object viewpoints. Thus, it would be beneficial if training with one transformation (*e.g.,* light changes) can improve performance on a different transformation (*e.g.,* viewpoint changes). We refer to such a regime as *assymetric diversity*—as models are trained with one kind of diversity, and tested on a different kind of diversity (**Fig. 5(e),(f)**). In all cases, the best generalization performance was obtained when training and testing with the same real-world transformation for both HVD (**Fig. 5(e)**) and Semantic-iLab datasets (**Fig. 5(f)**). In most cases, there was a drop in performance of $10\%$ or more when training in one transformation and testing with a different (assymetric) transformation. These experiments imply that to build models that generalize well, it is important to collect training data with multiple real-world transformations.

# F  ADDITIONAL EXPERIMENTS FOR THE ROLE OF CONTEXT

Given the success of HDNet, we asked whether implementing a two-stream separation of target and context would also improve performance for other architectures. We modified ResNet18 He et al. (2016) and ViT Dosovitskiy et al. (2020) to leverage scene context in the same way as HDNet. For ResNet, a two-stream version was made where each stream is a ResNet backbone. One stream operates on the target, and the other one on the scene context. Output features from each stream were concatenated, and passed through a fully connected layer for classification as shown in **Fig. 1**(c). The two-stream architecture for ViT was analogous. In contrast, the one-stream architecture did not use scene context and operated on the target object alone (see methods for additional details). The two-stream architectures consistently led to improved performance (two-sided t test, $p < 0.05$), as shown in **Table 3**.

To further understand the role of contextual information on visual recognition, we conducted two additional experiments. Firstly, we evaluated the impact of reducing scene context information by blurring it using a Gaussian Blur. As shown in **Table. 2**, performance dropped consistently for all three transformations as contextual information is reduced. Secondly, we confirmed that the increase in performance is due to the addition of contextual information and not due to the two-stream architecture *per se* by training HDNet with both streams receiving only the target information. This removal of context led to a drop in performance, as reported in **Table. S1** (see **Sec. F** for details).

Besides results on the role of context presented in **Table. 3**, we present here two additional experiments evaluating the contribution of scene context on generalization. Firstly, we also evaluated the impact of blurring the scene context while keeping the target intact Zhang et al. (2020). For each real-world transformation, we trained and tested models with increasing levels of Gaussian blurring applied to the scene context. These results are presented in Blurring was applied to the images in the form of a Gaussian kernel filter, with the kernel standard deviation ($\sigma$) set to 0, 25, or 125. The cropped image of the target object was passed to the second stream of the network without blurring. These results are reported in **Table 2**. As can be seen, there was a drop in performance as context blurred for all three real-world transformations.

| Semantic Shift | Target only | Target and Context |
|---|---|---|
| Viewpoint | 0.77 | **0.82** |
| Material | 0.85 | **0.94** |
| Lighting | 0.97 | **0.98** |

Table S1: **Training a two-stream HDNet with only target information**. As a third control for confirming the role of context, we train HDNet where both streams are passed just the target object. Thus, it is forced to learn without scene context. This results in a drop in performance for all semantic shifts, providing further evidence in support of the utility of scene context.

Secondly, we train HDNet such that both streams are trained with the target object. Thus, this modified version is forced to learn without scene context. These results are shown in **Table. S1**. For all semantic shifts, forcing HDNet to learn with only the target results in a drop in accuracy. This provides further evidence supporting the utility of scene context in enabling generalization.

# G  ADDITIONAL EXPERIMENTS WITH HDNET AND CONTRASTIVE LOSS

We evaluate the contribution of the contrastive loss by training variations of HDNet on HVD with and without the contrastive loss as shown in Eq. 2. These numbers are reported in **Table S2**. As can be seen, adding a contrastive loss improves performance for all three semantic shifts, providing evidence for its utility.

| Semantic Shift | Without Contrastive Loss | With Contrastive Loss |
|---|---|---|
| Viewpoint | 0.79 | **0.82** |
| Material | 0.89 | **0.94** |
| Lighting | 0.98 | **0.98** |

Table S2: **Impact of removing contrastive loss**. We evaluate the contribution of the contrastive loss by training and testing HDNet on the HVD dataset with and without the contrastive loss. The contrastive loss results in an improvement across all three semantic shifts.

| Test Dataset | ResNet | ViT | AND Mask | CAD | CORAL | ERM | IRM | MTL | Self Reg | VREx | HDNet (ours) |
|---|---|---|---|---|---|---|---|---|---|---|---|
| ScanNet | 0.35 | 0.29 | 0.43 | 0.40 | 0.42 | 0.48 | 0.46 | 0.46 | 0.53 | 0.42 | **0.61** |

Table S3: **Human visual diet improves generalization to larger real world dataset as well**. We curated a larger subset of ScanNet images, allowing more complex real world scenarios like blurry images, clutter and occlusions. We report the capability of models to generalize from synthetic HVD images to this more complex subset of ScanNet. HDNet leveraging human-like visual-diet outperforms all baselines on this more complex dataset as well.

## H  ADDITIONAL EXPERIMENTS WITH A LARGER, LESS CONTROLLED SCANNET TEST SET.

We extend the generalization to real-world results presented in the main paper by reporting these numbers on a larger test set created by annotating additional images from ScanNet. As ScanNet was created by shooting video footage of 3D scenes, many frames can be blurry. In the original, smaller test-set such blurry frames were removed to ensure a higher quality test set. However, here we also include additional images with lower fidelity to report numbers on a larger test set. These numbers are reported in **Table. S3**. The trend is consistent with results reported on a smaller, more controlled subset in the main paper—HDNet outperforms all other benchmarks by a large margin. As expected, including these images in the test set results in a drop in accuracy across all methods. All models were trained on synthetic images from HVD and were tested on a test set of natural images from ScanNet.

## I  HYPERPARAMETERS

**HDNet:** As our model builds on top of CRTNet Bomatter et al. (2021) as backbone, we use the same hyperparameters for the backbone as reported in the original paper. All models were trained for 20 epochs with a learning rate of 0.0001, with a batch size of 15 on a Tesla V100 16Gb GPU.

**Domain generalization:** We used the code from Gulrajani et al. Gulrajani & Lopez-Paz (2020) to train and test domain generalization methods on our dataset. The code is available here: `https://github.com/facebookresearch/DomainBed`. To begin, we ran all available models and tried 10 random hyperparameter initializations. Of these, we picked the best performing hyperparameter seed—24596. We also picked the top performing algorithms as the baselines reported in the paper.

**FasterRCNN:** We used the code from Bomatter et al. Bomatter et al. (2021) to train and test the modified FasterRCNN model for recognition. The code is available here: `https://github.com/kreimanlab/WhenPigsFlyContext`, and we used the exact hyperparameters mentioned in the repository.

## J  EXPERIMENTAL DETAILS

HDNet was compared against several baselines presented below. All models were trained on NVIDIA Tesla V100 16G GPUs. Optimal hyper-parameters for benchmarks were identified using random search, and all hyper-parameters are available in the supplement in **Sec. I**.

### J.1  BASELINE APPROACHES

We compared the impact of a human-like visual diet with a diverse set of alternative approaches popular in machine learning. This includes:

**2D feed-forward object recognition networks:** Previous works have tested popular object recognition models in generalization tests Geirhos et al. (2018); Boyd et al. (2022). We include the same popular architectures ranging from 2D-ConvNets to transformers: DenseNet Iandola et al. (2014), ResNet He et al. (2016), and ViT Dosovitskiy et al. (2020). These models do not use context, and take the target object patch $I_t$ as input.

**Domain generalization methods:** We also compare HDNet to an array of state-of-the-art domain generalization methods (**Table 1**). These methods also use only the target object, and do not use contextual information.

**Context-aware recognition models:** To compare against models which use scene context, we include CRTNet Bomatter et al. (2021) and Faster R-CNN Ren et al. (2015). CRTNet fuses object and contextual information with a cross-attention transformer to reason about the class label of the target object. We also compare HDNet with a Faster R-CNN Ren et al. (2015) model modified to perform recognition by replacing the region proposal network with the ground truth location of the target object.

**Billion-Scale self and semi supervised architectures:** We presented results with a suite of modern approaches trained on 1000-fold more data to emphasize the importance of data quality over sheer dataset size. These included—Dino V2, ResNet50 SWSL, ResNet18 SWSL, 32x4d SWSL, ResNext101 32x16d SWSL, and ResNext50 32x4d SWSL.

### J.2  EVALUATION OF COMPUTATIONAL MODELS

Performance for all models is evaluated as the Top-1 classification accuracy. Error bars reported on all figures refer to the variance of per-class accuracies of different models. For statistical testing, p-values were calculated using a two-sample paired t-test on the per-category accuracies for different models. The t-test checks for the null hypothesis that these two independent samples have identical average (expected) values. For ScanNet, a t-test is not optimal due to the smaller number of samples, and thus a Wilcoxon rank-sum test was employed for hypothesis testing as suggested in past works De Winter (2019); Posten (1982). All statistical testing was conducting using the python package *scipy*, and the threshold for statistical significance was set at 0.05.

