# OpenReview forum: "Improving out-of-distribution generalization by mimicking the human visual diet."
_ICLR.cc/2025/Conference — ICLR 2025 Conference Withdrawn Submission_

### Official Review · Reviewer_vYhT · 2024-11-02

**Soundness:** 2
**Presentation:** 3
**Contribution:** 2
**Rating:** 5
**Confidence:** 3

**Summary:**

The paper introduces a novel idea to improve OOD generalization by training models on a dataset that mimics the attributes of a human-like visual diet, focusing on real-world transformational diversity (RWTD) and scene context. To achieve this end, a Human Visual Diet(HVD) dataset is proposed, and a new model(Human Diet Network) is designed to leverage the attributes of a human-like visual diet.  The authors have demonstrated that incorporating  visual diet attributes into training data significantly improves generalization to OOD changes in lighting, materials, and viewpoints.

**Strengths:**

1. The paper proposes a novel perspective that mimicking attributes of the human-like visual
 diet improves the generalization to OOD lighting, material, and view point changes.

2. A new dataset, Human Visual Diet (HVD), and a new model, Human Diet Network (HDNet), have been introduced to better  leverage the characteristics of human vision.

3. The paper comprehensively evaluates the performance of HDNet via experiments on multiple datasets and the improvements in OOD generalization is significant.

**Weaknesses:**

1. Scene context is extracted with a network. When training this network, is it a binary classification task to distinguish "scene" and "non-scene"? This should be further explained.

2. What is real-world transformation in the training? More specific samples are needed for further understanding. It seems that this paper improves the OOD to view points changes via training on more samples with different view point changes. If so, the training and testing are independently and identically distributed. It sounds lack of novelty.

3. The core idea of HDNet is Contrastive Learning. Many methods based on Contrastive Learning are proposed to improve OOD. This paper should compare with these methods as well.

**Questions:**

See the  Weaknesses above.

---

### Official Review · Reviewer_65JY · 2024-11-03

**Soundness:** 2
**Presentation:** 3
**Contribution:** 2
**Rating:** 3
**Confidence:** 4

**Summary:**

The paper makes two contributions: 1) the Human Visual Diet Dataset (HVD), a 300k image synthetic datasets of 15 high-quality rendered environments (using OpenRoom), exhibiting large amount of lighting, material, and viewpoint changes.   2) Visual Diet Network, a two-stream CNN/transformer object/context architecture.  The VDNet takes in an image and also cropped version of the object, and outputs the object's class.  Additional contrastive losses are added for viewpoint, lighting, and material invariance.
Extensive experiments demonstrate that VDNet performs well on HVD dataset compared to generic baselines.

**Strengths:**

The paper is focused on a very important topic -- the important of data quality, not just quantity -- for performing real-world tasks.   The abstract and intro argues beautifully that the most popularly used training visual datasets (millions of random images unconnected with each other) are very far away from the typical visual diet that a child is exposed to growing up (a small number of environments, think a single house, but huge number of different conditions within that entombment).  Indeed, if the rest of the paper just focused on the data side of the story (using standard, SOTA networks), it would be a fantastic contribution.

**Weaknesses:**

Unfortunately, while the abstract/intro is all about the data, the rest of the paper is mainly about the VDNet.   Most of the experiments presented conflate the dataset and VDNet, so so it is impossible to know if the results are due to the data, the network, or indeed the task.   Indeed, in many cases, the experiments are "doomed to succeed", because VDNet gets an unfair advantage over standard networks by seeing more of the image, and by knowing about transformations.  For example, in 5.2, VDNet is trained on  HVD dataset, but baselines seem to be trained on iLab dataset.  Clearly, since VDNet has the contrastive term, it should better handle various appearance changes, even OOD ones that it wasn't specifically trained for, than a standard network trained on classification alone.

Other issues:
" The paper makes it sound that internet-scaped images do not include context, only cropped objects.  This is certainly not true -- there has been a long line of work on understanding scenes, PASCAL, LabelMe, COCO,, SUN, etc etc.
* The way context is modeled here seems very unrealistic -- the model is given the bounding box of the object plus the full image.  So, in a sense, the detection problem has already be solved, on the classification problem remains.  It's hard to see how could this happen in the real world, where detection and classification must happen together.
* Related work is missing a discussion of a whole line of work that also aims to give the models a better "Visual Diet" --  Videos, and, in particular, Egocentric videos.  A lot of the goals of HVD (fewer settings, more variation) are shared by efforts like EpicKitchens, Ego4D, KrishnaCam, as well as:
https://deepmind.google/research/publications/59160/
* Line 406: Huang & Belongie, 2017 is claimed to be "Generative AI" way before this term was even in wide use.  To make this claim, a much more contemporary approach should be used.

**Questions:**

* Interestingly, researchers have been finding that training models like DINO on egocentric video like Ego4D (from scratch) does not work as well as training on ImageNet.  This result is not consistent with arguments in the paper.  It would be interesting to think if the problem is the particular data, or the model/task.

* Since DVNet has been pretrained with ImageNet 1 million images, it is not really following the Human Visual Diet as promised.  What happens if you train DVNet only on HVD from scratch?

---

### Official Review · Reviewer_VkYx · 2024-11-04

**Soundness:** 3
**Presentation:** 2
**Contribution:** 2
**Rating:** 3
**Confidence:** 4

**Summary:**

This paper introduces a synthetic setting to study model generalization. Using reconstructions of ScanNet scenes, synthetic images are generated by modifying lighting, materials, and camera viewpoint. Models are evaluated by how well they generalize to new lighting, new materials, or new viewpoints.

The problem setting is object classification where the input is an image as well as a bounding box to define the object location. A two-stream network (Bomatter 2021) is used that takes in two images: 1) the full input image and 2) a crop to the provided box. The output is an ensembled prediction from the two network streams. This is supervised with both a classification and contrastive loss.

The authors compare against various baseline models, most of which are only provided the cropped bounding box input. These fare worse than the proposed network. Additionally, comparisons are made when training on a less diverse version of the dataset. Both the full scene context and more data diversity are important to achieving strong results.

**Strengths:**

The authors perform a wide variety of experiments across many settings to control for different factors that might influence performance:
- we see that models are less adversely affected by novel lighting than materials or viewpoints, and get very convincing results that the additional input stream providing full image context is important for good performance
- some interesting baselines for adding image diversity (data augmentation and image style transfer) are shown to be less helpful than more diverse synthetic data

**Weaknesses:**

It is not obvious to me that today's datasets are missing substantial viewpoint, material and lighting variation? This paper shows that training on a version of a synthetic dataset with more material and viewpoint changes yields better generalization than a version with less variation, which is great, but it is unclear what takeaway there is about how any modern large scale internet dataset is insufficient.

It seems that by the nature of this particular benchmark, there is no way for a model to do well that only receives a cropped input (judging by the results in Table 3). But most of the comparisons to other networks only provide a cropped input which renders those comparisons less interesting, should we expect any of the models in Table 4 to be competitive? Similarly, the contrastive loss seems to be helpful (Appendix G), are any of the baselines trained with it? There's a number of places where overall it is not clear what is being compared in an apples-to-apples way (e.g. what is being controlled for in terms of data vs architecture vs losses).

Some aspects of the presentation suggest the submission might need more polish and time before it is at an appropriate level for publication:
- overall the writing and organization are not as clear as they could be and crucial details for the reader are sometimes missing. For example, I do not have a sense of how the proposed HDNet differs from CRTNet? Is it the contrastive loss or are there architectural changes?
- I think CRTNet is missing from Table 1?
- Some of the writing in the appendices feels a bit redundant with the writing in the main paper and there's an appendix section that is empty (Appendix D)

**Questions:**

The train/test differentiation for these experiments is not entirely clear to me, are scenes reused in both settings? For example, to report the results on "Lighting", does this mean the network was already trained on samples with identical objects with the same materials and same camera viewpoint but now with new lighting?

I do think the difference in distribution between internet training images and the "human visual diet" is an interesting one. Have the authors considered comparing to a data setting like Ego-Exo4D which provides egocentric data that might be more aligned with our "visual diet" than say, a typical internet-scraped dataset?

---

### Official Review · Reviewer_nsov · 2024-11-04

**Soundness:** 3
**Presentation:** 2
**Contribution:** 3
**Rating:** 5
**Confidence:** 4

**Summary:**

This paper investigated how the differences in input data composition (i.e., visual diet) impact the Out-Of-Distribution (OOD) generalization capabilities of a visual system. In conclusion, training models on a dataset mimicking attributes of the human-like visual diet improved generalization to OOD lighting, material, and viewpoint changes by up to 18%. This observation held despite the fact that the models were trained on 1, 000-fold less training data. Furthermore, when trained on purely synthetic data and tested on natural images, incorporating these visual diet attributes in the training dataset improved OOD generalization by 17%. These experiments are enabled by our newly proposed benchmark—the Human Visual Diet (HVD) dataset, and a new model (Human Diet Network) designed to leverage the attributes of a human-like visual diet. These findings highlight a critical problem in modern-day Artificial Intelligence—building better datasets requires thinking beyond dataset size and rather focus on improving data composition. All data and source code will be made available upon publication.

**Strengths:**

1. This paper constructs a new dataset with a clear motivation --- to mimic the human visual diet in real-world scenarios.

2. The model trained on this dataset does demonstrate the superiority of real-world deployment, demonstrating the effectiveness of the dataset.

**Weaknesses:**

1. I am satisfied with the construction and motivation of the dataset. However, relatively speaking, the design of the network structure is a bit unimpressive. While the proposed dataset demonstrates novelty and effectiveness, this paper falls short in showcasing how to fully leverage its potential. Rather than proposing novel techniques tailored to this dataset, the authors primarily apply conventional methods, such as contrastive learning. This design appears somewhat conventional given the unique attributes of the dataset. To better explore the dataset’s potential, a more in-depth theoretical analysis should be conducted to clarify the relationship between the dataset and the network design, illustrating how this approach can truly harness the value of human visual diet-aligned data.

2. This paper claims to align biological and artificial vision systems; however, it addresses only three domain gaps deemed important by the authors without providing supporting biological analysis. I encourage the authors to conduct a more in-depth analysis to substantiate this claim and enhance the rigor of their approach.

Minor issues:
There are quite a few typos in the manuscript. Such as "Fist" -> "First";

**Questions:**

Please see the "Weaknesses" part.

---

> ### Comment · Reviewer_nsov · 2024-11-29
>
> There hasn't been any discussion of the problems I raised. Considering the comments of other reviewers, I temporarily reduced the score to 5.

---

### Note · Authors · 2024-12-03

**Comment:**

We thank the reviewers for their feedback and valuable suggestions. Based on the feedback, we have decided to withdraw the manuscript. Thank you.

**Withdrawal Confirmation:**

I have read and agree with the venue's withdrawal policy on behalf of myself and my co-authors.